# Management of Polydrug-Resistant Tuberculosis

**DOI:** 10.3390/medicina59020246

**Published:** 2023-01-27

**Authors:** Mediha Gonenc Ortakoylu, Isil Kibar Akilli, Lutfiye Kilic, Belma Akbaba Bagci, Esma Seda Akalin, Arzu Deniz Aksan, Sezer Toprak, Mehdi Mirsaeidi

**Affiliations:** 1Department of Pulmonary Medicine, Yedikule Chest Diseases and Thoracic Surgery Training and Research Hospital, University of Health Sciences Turkey, Istanbul 34020, Turkey; 2Department of Pulmonary Disease, Dr. Sadi Konuk Training and Research Hospital, University of Health Sciences Turkey, Istanbul 34149, Turkey; 3Department of Pulmonary Medicine, Koç University Hospital, Koç University, Istanbul 34010, Turkey; 4Department of Microbiology, Yedikule Chest Diseases and Thoracic Surgery Training and Research Hospital, University of Health Sciences Turkey, Istanbul 34020, Turkey; 5School of Medicine, University of Florida, Jacksonville, FL 32611, USA

**Keywords:** tuberculosis, drug resistance, polydrug-resistant tuberculosis, diagnosis, treatment, management

## Abstract

*Background and Objectives*: There is a lack of information regarding the effective duration of treatment necessary to prevent the development of acquired resistance when fluoroquinolones (FQ), and/or pyrazinamide (Z) resistance has occurred in patients with polydrug-resistant tuberculosis and isoniazid resistance. The management of these kinds of patients should be carried out in experienced centers according to drug susceptibility test results, clinical status of the patient and the extensity of the disease. *Materials and Methods*: We evaluated treatment regimens, treatment outcomes, and drug adverse effects in seven patients with polydrug-resistant tuberculosis, including those with Z and/or FQ resistance in a retrospective analysis *Results*: Regarding the patients with polydrug-resistant tuberculosis in addition to isoniazid (H) resistance, three had Z, two had FQ, and the remaining two had both Z and FQ resistance. In the intensive phase of the treatment, the patients were given at least four drugs according to drug susceptibility tests, and at least three drugs in the continuation phase. The duration of treatment was 9–12 months. Two of the patients were foreign nationals, and could not be followed up with due to returning to their home countries. Regarding the remaining five patients, three of them were terminated as they completed treatment, and two as cured. No recurrence was observed in the first year of the treatment. The most common, and serious drug side effect was seen for amikacin. *Conclusions*: In patients with polydrug-resistant TB, if Z and/or FQ resistance is detected in addition to H resistance, the treatment of these patients should be conducted on a case-by-case basis, taking into account the patient’s resistance pattern, clinical condition, and disease prognosis. Close monitoring of the side effects will increase the success rate of the treatment.

## 1. Introduction

It was reported that over 10.6 million people were diagnosed with Tuberculosis (TB) in 2022. This number has remained relatively stable over the past few years, but drug-resistant TB continues to present a major risk for public health. Among all recorded TB cases, there are approximately half a million rifampicin-resistant TB (RR-TB) cases, and 78% of these are multidrug-resistant TB (MDR-TB) cases [1]. Resistance to isoniazid is also quite common worldwide. Isoniazid is an important first-line anti-TB drug due to its strong early bactericidal activity against *M. tuberculosis* [2]. The ratio of isoniazid-resistant, rifampicin-susceptible TB (Hr-TB) was 7.2% in new TB cases, and 11.6% in previously treated TB cases [3].

In the WHO 2022 report, the incidence of tuberculosis in Turkey is 18 cases per 100,000 people. Turkey is among the countries with a moderate to low incidence of TB, where 2.4% of new cases and 8% of previously treated cases were reported to be MDR/RR tuberculosis (1).

With respect to a systematic review and meta-analysis comparing the treatment outcomes of Hr-TB and Drug-Susceptible TB, it was reported that the results of Hr-TB treatment with first-line drugs were suboptimal with higher failure rates (11% vs. 1%) and relapse rates (10% vs. 5%) [4]. Furthermore, a recently published meta-analysis study that compared different treatments of Hr-TB by adding fluoroquinolone to the 6-month daily rifampicin, ethambutol, pyrazinamide (REZ) regimen showed to provide optimal results in the treatment of Hr-TB, whereas adding isoniazid to the treatment and extending the daily REZ treatment for more than six months did not provide any benefit [5].

Pyrazinamide and fluoroquinolones are included in treatments as important components of treatment regimens for MDR/RR, and isoniazid-resistant, rifampicin-susceptible TB [6]. Regarding the data from levofloxacin and pyrazinamide resistance studies conducted among tuberculosis patients in six countries among Hr-TB patients, levofloxacin resistance was found to be 0.0–13.5%, pyrazinamide resistance 0.0–8.7%, and resistance to both drugs was 0.0–9.5% [7].

TB infections that occur due to organisms demonstrating in vitro drug resistance to more than one anti-TB drug (without R) are referred to as polydrug-resistant. Any number of combinations of resistance can occur [8].

There is currently no evidence-based recommendation regarding an effective regimen for treatment, and prevention of the development of acquired resistance when FQ and/or Z resistance occur in Hr-TB patients. The management of such cases is carried out on a case-by-case basis by centers (clinicians) experienced in using second-generation drugs, with consideration of the patient’s resistance pattern, clinical condition, and disease prognosis. [6,9].

Here, we present a case series of patients with Hr-TB and with Z or FQ or both Z and FQ resistance in our center and evaluate their clinical outcomes.

## 2. Materials and Methods

The present retrospective study was conducted in the Yedikule Chest Diseases and Thoracic Surgery Training and Research Hospital. This hospital is one of the four national referral hospitals designated for TB in Turkey. The study was approved by the Institutional Ethics Committee of Yedikule Chest Diseases and Thoracic Surgery Training and Research Hospital. Written informed consent was obtained from each patient participating in the study.

Seven patients who were hospitalized with Hr TB and determined to have Z or FQ or both Z and FQ resistant strains were included in this study. Two of the patients were born and lived in foreign countries with high resistance rates against anti-TB drugs (Kazakhstan and Azerbaijan). Following Turkey’s National TB Control Program guidelines, three sputum samples were collected from each patient at the diagnosis, and during each month of the treatment [10]. Sputum smears and cultures were examined weekly to detect early responses to treatment in this difficult-to-treat patient group.

Genotypic Drug-Susceptibility Testing (DST) was performed for rifampicin and isoniazid on each sputum-smear positive patient’s samples in same center; resistance to quinolone of another injectable agent was also tested in case of detecting MDR-TB or detecting Hr-TB and planning to add quinolone to the treatment. Respiratory samples were cultivated in both liquid and solid (Lowenstein–Jensen) media [11]. When reproduction occurred within a culture, a Phenotypic Drug Susceptibility test was performed to confirm the results of the Genotypic Resistance Test. Phenotypic DST was applied to each sample that was negative in the smear and positive in the culture.

Phenotypic Drug-Susceptibility Testing: We used the standard protocol for DST in the MGIT 960 (Becton Dickinson Diagnostic Instruments, Sparks, MD, USA) following the manufacturer’s instructions. The final critical concentrations were 0.1 μg/mL for isoniazid, 1.0 μg/mL for rifampicin, 5.0 μg/mL for ethambutol, and 2.0 μg/mL for streptomycin [12]. Additionally, a phenotypic susceptibility test to pyrazinamide was conducted on the MGIT 960 at 100.0 μg/mL using an MGIT-PZA kit for moxifloxacin, whereas the concentration used for levofloxacin was 1.5 μg/mL [13].

Genotypic Drug-Susceptibility Testing: The real-time PCR method (Anyplex MTB/NTM Real-time detection, Seegene) was employed for the MTB/NTM distinction of isolated DNA. MTB-positive samples were tested for Isoniazid for MDR, four mutations for katG, three mutations for inhA promoter, and 18 mutations for the rifampicin rpoB gene. Besides, the following tests were performed: Fluoroquinolones (FQ) for extensively drug-resistant tuberculosis (XDR-TB); seven mutations for gryA and injectable aminoglycosides (amikacin, Kanamycin, Capreomycin), rrs gene, and six mutations for the eis promoter gene region (Anyplex II MTB/MDR/XDR Detection, Seegene) [14]. 

Treatment regimens were tailored based on the DST results. Clinical outcomes were defined as completed treatment, and cure as previously suggested [15]. If two consecutive cultures were negative, the first negative culture’s date was recorded as the time of culture negativity.

## 3. Results

Two out of the seven patients were born and raised in Kazakhstan and Azerbaijan, which have high resistance rates against anti-TB drugs. All patients were HIV-negative, and there were no known index cases. The age of the patients was between 34 and 69, with an average of 48.3 ± 12.6. Only one patient was female (case 3), and she was from Kazakhstan.

Two of the seven patients had previously received TB treatment. One patient (case 4) was born in a foreign country and was treated for TB 3 years prior to arrival. However, we were unable to obtain clinical information concerning their treatment regimen. The other patient (case 6) had used 2HRZEFQ + 4HREFQ for six months after detecting H and E resistance in the diagnosis. However, quinolone resistance was not investigated in this period.

The demographic characteristics and resistance test results of the patients are shown in Table 1.

Bacteriological follow-up could not be performed with three of the patients: one with a positive pleural fluid culture, another with a positive bronchoscopy lavage culture, and one that could not produce sputum. Time to culture negativity of the patients was (20–70) days on average 33.8 ± 20.4. Three of the polydrug-resistant TB patients had Z, two had FQ, and two had both Z and FQ resistance. The patients’ treatment regimens were arranged with at least four drugs in the intensive phase, which were effective according to the resistance results, and with at least three drugs in the maintenance phase.

The patients’ resistance patterns, the treatment regimens used, treatment durations, drug side effects, and treatment results are shown in Table 2.

Cases 3 and 4 were discharged from the hospital on the 57th and 59th days of the treatment and returned back to their countries. Clinical, and radiological improvement was detected in both patients; case 4 became negative culture on the 20th day of the treatment, and the other patient could not give a sample. Two patients had side effects that required the discontinuation of three drugs. In case 4, hearing loss developed on the 43rd day of the treatment, and despite the withdrawal of amikacin from the treatment and intratympanic steroid therapy, the hearing loss was permanent. This treatment was applied by an otorhinolaryngologist considering that the rapidly developing hearing loss might be partially reversible [16]. In case 6, cycloserine was discontinued due to orientation disorder development on the 12th day of the treatment, and amikacin was discontinued due to hearing loss on the 30th day.

Treatment results of seven patients were reported as follows; two patients as not evaluated (returned to their home country), two patients as completion of treatment, and three patients as cured. The duration of treatments was planned as nine months, and the treatment of the patient with FQ and Z resistance (case 7) continued for 12 months. No recurrence was detected in the first-year follow-up of the five patients who could be followed. Regarding Case 4’s chest X-ray images at the beginning and the 59th day of the treatment (Figure 1 and Figure 2) and Case 7’s chest X-ray images at the beginning and the 12th month of the treatment (Figure 3 and Figure 4) a significant radiological improvement was observed.

## 4. Discussion

In our study, five of seven polydrug-resistant tuberculosis patients with isoniazid resistance plus pyrazinamide and/or fluoroquinolone resistance were successfully treated. Two patients could not be followed due to transfer, but bacteriological, clinical and radiological responses occurred in the second month of the treatment. Despite the small number of cases, our study provides concrete results to guide clinicians for treating similar patients.

The basic principle of treatment success in resistant tuberculosis is the early diagnosis of drug resistance and the establishment of a curative treatment regimen according to the patient’s DST results [17]. Rifampicin is the most important drug in the short-term treatment regimen. Since the treatment success is very low in R-resistant cases, a rapid test for R resistance is recommended at the beginning of treatment. This recommendation has taken its place in the guideline. It is recommended to treat RR patients like MDR-TB patients [18,19].

Isoniazid-resistant, rifampin susceptible tuberculosis (Hr-TB) is globally the most common form of drug-resistant TB [3]. Isoniazid resistance has been known for many years for its high bactericidal activity and its ability to prevent acquired resistance development. The effect of isoniazid resistance on the treatment outcomes seems to have been somehow ignored. In the study researching the effect of initial H resistance on MDR-TB development, most of the acquired MDR development has been shown to occur in initial H-resistant cases; an accurate diagnosis and tailored treatment of Hr-TB have been shown to reduce MDR development by almost 50%. It has been emphasized that constant adherence to diagnosis and treatment protocols ignoring Hr-TB will likely result in more MDR-TB cases [20].

The success of the 6RZEFQ regimen administered to Hr-TB patients has been demonstrated. In this regimen, it is important to know the resistance status against Rifampicin-sparing essential anti-TB drugs, pyrazinamide, and fluoroquinolones. Treatment becomes difficult in the presence of drug resistance other than R; it should be managed similarly to MRD-TB treatment in private centers, with second-generation drugs when necessary [9]. Although some suggestions are made for R-susceptible polydrug-resistant cases in Turkey’s National TB Control Program guidelines, structuring and following the treatment on personal bases is recommended (10).

We can outline our experiences regarding the treatment protocols administered to the seven patients in our study. The treatment’s intensive phase should be carried out with at least four drugs and the continuation phase with at least three drugs. In the treatment, levofloxacin was chosen as the basic FQ drug in addition to rifampicin. Levofloxacin (Lfx) was preferred because published studies showed a better safety profile for Lfx and rifampicin decreased the plasma peak concentration of moxifloxacin but Lfx did not have such an interaction [21,22].

In our study, drugs found to be resistant in DST were not included in the regimen even though the patient had not used them before; thus, drug side effects were avoided [23]. The most serious side effect was observed in amikacin, causing permanent damage even in short-term use (30 days and 43 days). In 2018, it was reported that kanamycin and capreomycin, which are the injectable drugs used in the treatment of MDR, should not be used and it was appropriate to use amikacin only in more critical situations than group C. Thus, the importance of the recommendation made in the guideline regarding injectable drugs was emphasized again [24].

Studies have shown that the sputum culture’s time to negativity is a good prognostic factor showing MDR-TB treatment success [25,26]. We closely followed the sputum smear and cultures in our patient group. We want to underline again that sputum smear and culture tests are good indicators of treatment response. Regarding this patient group, where there are no precise data on the duration of the treatment, it would be appropriate to determine the duration of treatment as 9–12 months, considering the resistance pattern, the drugs in the regimen, the extent of the disease, and the time to culture negativity.

The current case report is limited due to the small number of cases and the failure to follow the two cases’ treatment outcomes. Regarding four cases with genotypic FQ resistance, FQ resistance was shown phenotypically in two cases, whereas phenotypic resistance could not be examined in the other two. Z resistance was studied only phenotypically in all cases. In addition, two anti-TB drugs, bedaquilin and delemanid, which were introduced in 2013 and are gradually becoming more commonly used in international practices, are still used only in XDR-TB patients in our country [19]. There is a need for studies on the use of these drugs in difficult-to-manage polydrug-resistant TB patients. Pyrazinamide or levofloxacin resistance is higher among Hr-TB patients than rifampicin and isoniazid-sensitive patients, but combined resistance to pyrazinamide and levofloxacin is rare [7]. Therefore, it does not seem possible to conduct randomized controlled studies consisting of patients with such resistance patterns.

## 5. Conclusions

Our study offers suggestions that clinicians can use in their clinical practice. In our case series with seven patients with polydrug-resistant TB, performing the intensive phase of the treatment regimen with four or five effective drugs and using at least three drugs in the continuation phase increased our treatment success. We would like to emphasize that the duration of treatment should be at least 9 months and also close bacteriological monitoring of the treatment response of the patients is important. The success in treating this patient group is mostly based on accurate diagnosis and DST results. The drugs’ side effects should be managed carefully, and the duration of the treatment should be determined according to the patient’s response to the treatment. We think that testing H resistance at the beginning of the treatment is as important as testing R resistance is the basis of success in tuberculosis treatment. When H resistance is detected, the rapid recognition of Z and FQ resistance will prevent the emergence of MDR and even XDR, which will increase the success of the treatment.

## Figures and Tables

**Figure 1 medicina-59-00246-f001:**
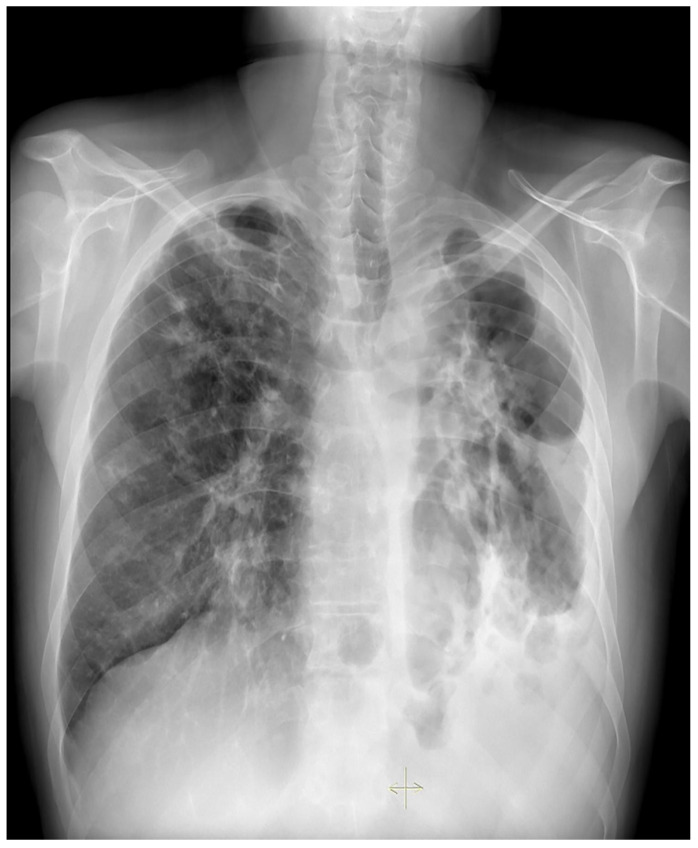
Case 4, beginning of the treatment.

**Figure 2 medicina-59-00246-f002:**
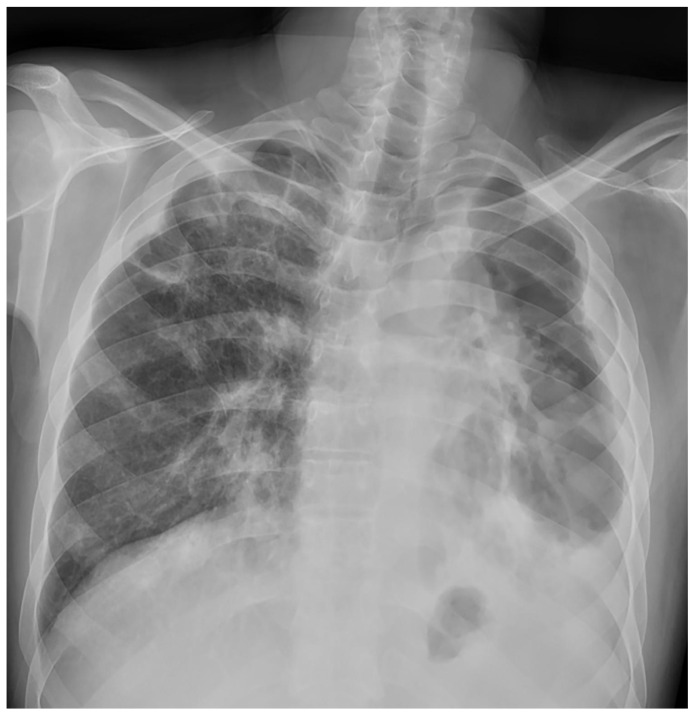
Case 4, 59th day of the treatment.

**Figure 3 medicina-59-00246-f003:**
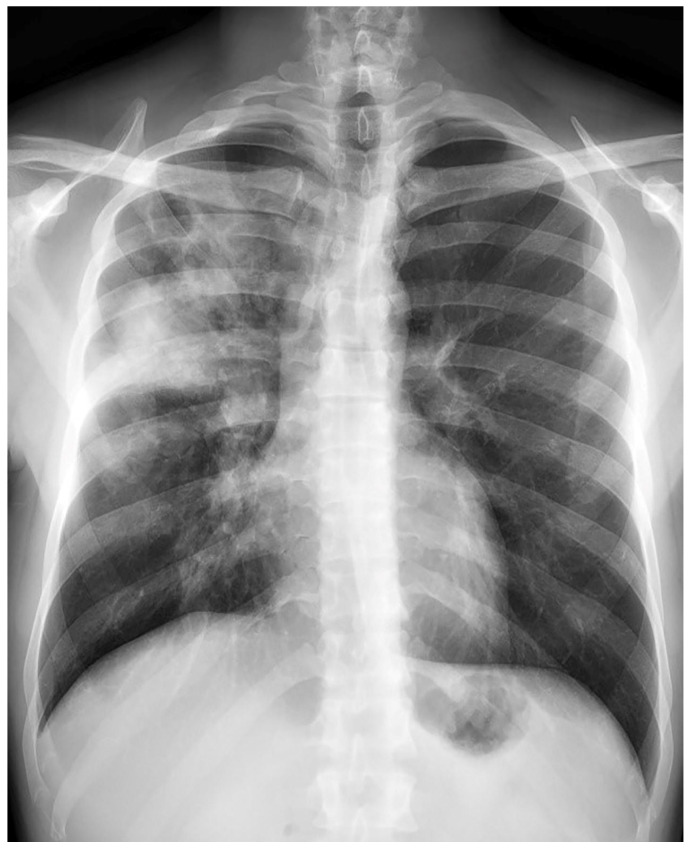
Case 7, beginning of the treatment.

**Figure 4 medicina-59-00246-f004:**
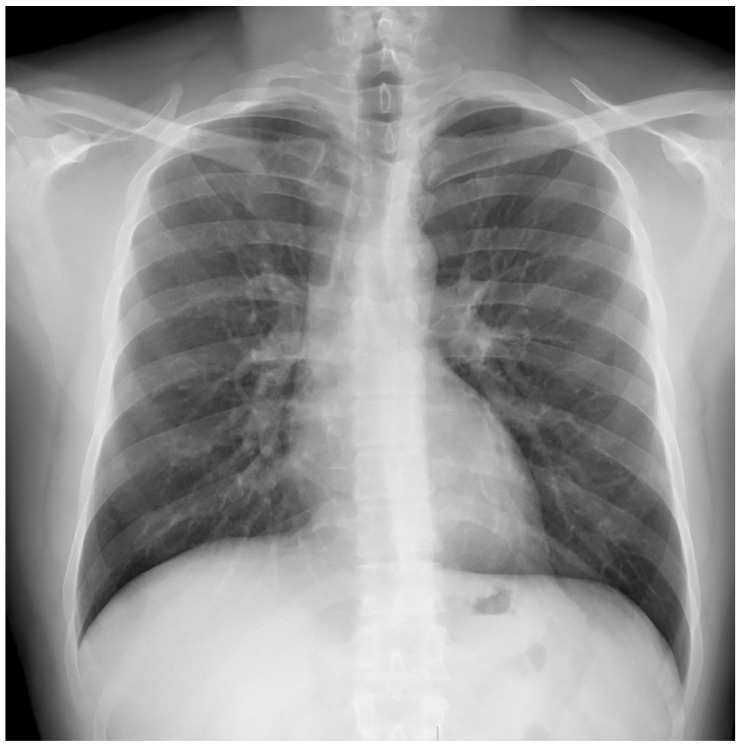
Case 7, 12th month of the treatment.

**Table 1 medicina-59-00246-t001:** Characteristics of the polydrug- resistant TB patients.

	Case 1	Case 2	Case 3	Case 4	Case 5	Case 6	Case 7
Sex	Male	Male	Female	Male	Male	Male	Male
Age (years)	57	48	44	34	52	69	34
Origin	Turkey	Turkey	Kazakhstan	Azerbaijani	Turkey	Turkey	Turkey
Treatment history	None	None	None	Yes	None	Yes	None
Site of TB	Lung	Lung	Lung	Lung	Pleura	Lung	Lung
Cavite	Yes, bilateral	Yes	None	Yes, bilateral	None	None	Yes
Co-morbidity	DM	None	RA	Pemphigus	DM and IHD	IHD	None

Abbreviations: DM, Diabetes Mellitus; RA, Rheumatoid Arthritis; IHD, Ischemic Heart Disease.

**Table 2 medicina-59-00246-t002:** Drug susceptibility testing results, applied treatment regimens, and outcomes of the polydrug-resistant TB patients.

	Case 1	Case 2	Case 3	Case 4	Case 5	Case 6	Case 7
Bacteriology	Smear +	Culture +	Culture +	Smear +	Culture +	Smear +	Smear +
Resistance pattern	H, Z, S	H, Z, E, S	H, Z, S	H, Z, E, S, FQ	H, E, S, FQ	H, E, S, FQ	H, Z, S, FQ
Previous treatment	80-day HRZE	None	None	Unknown	None	150-day H, R, Z, E, Mox	47-day R, E, Z, Mox
Treatment regimen intensive phase	3R, E, Lfx, Cs, Am	5R, Lfx, Cs, PAS, Am	R, E, Lfx, Cs, Pto, Am	R, Lzn, Pto, Cs, Am, Z *	9R, Z, Cs, Pto	1R, Z, Lzn, Pto **, PAS **, Cs, Am	4R, E, Cs, Pto, Am
Continuation phase	6R, E, Lfx	4R, Lfx, Cs, PAS	Not known	Not known	Nine months with the same regimen	8R, Z, Lzn, Pto, PAS,	8R, E, Cs, Pto
Total period	9 months	9 months			9 months	9 months	12 months
Time to culture negativity	26th day	26th day	Could not produce sputum	20th day	Could not be tested	70th day	27th day
Treatment Outcome	Cure	Completion of treatment	Not evaluated	Not evaluated	Completion of treatment	Cure	Cure
Drug side effect	None	None	None	Hearing loss	None	Orientation disorder Hearing loss	None
Discontinued drug				Amikacin (Used for 43 days)		Cs (Used for 12 days) Amikacin (Used for 30 days)	

Abbreviations: H, Isoniazid; R, Rifampicin, Z, Pyrazinamide; E; Ethambutol; S; Streptomycin; Mox, Moxiflixacin, Lfx, Levofloxacin; Cs, Cycloserin; Pto, Protionamid; Am, Amicasin; Lzn, Linezolid, PAS, Para aminosalicylic acid, * Z Used 21 days until DST result came out, ** Added due to discontinuing Cs and Am.

## Data Availability

The data presented in this study are available on request from the corresponding author.

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
