# Peer review of "Management of Polydrug-Resistant Tuberculosis"

_medicina, 2023, doi:10.3390/medicina59020246_

Round 1

Reviewer 1 Report

This paper reviews the outcomes of 7 patients with tuberculosis that is resistant to isoniazid and either pyrazinamide, fluroquinolones or ethambutol.

Two cases to do have outcomes.

Comments

1. The cases were seen at a national referral centre and were all admitted. Are these cases typical of drug resistant cases and if not a comparison could be made.

2. I could not see a discussion of drug resistance data from other centres in Turkey or nationally this should be included. 

3. How was treatment monitoried as an inpatient and as an outpatient?

4. The x-rays presented are cut off

5. Were there treatments given in line with national guidelines?

6. Are there historical patients to compare outcomes to?

Author Response

Thank you for evaluating our manuscript. Thank you for your contribution and interest.

Changes were made based on your suggestions.

A native speaker has reviewed and corrected the text.

1) Although rifampicin susceptible, it is not possible to make a comparison due to the small number of cases with pyrazinamide and/or quinolone resistance in addition to isoniazid, which has an important place in the treatment.

2) Turkey's data on TB in the 2022 WHO Tuberculosis Report were cited in the manuscript.

3) Turkey's National Tuberculosis Control Program has a vertical structure. Within this structure, all follow-ups and treatments are carried out in cooperation from the periphery to the center.

4) Chest X ray images were corrected according to your recommendations.

5) Were there treatments given in line with national guidelines? The treatmens given were mentioned in the discussion section of the article as you recommended.

6) We think that all over the world there are definitely cases similar to the cases we have presented, but unfortunately we have not come across any study in which such cases are presented.

Reviewer 2 Report

Extensive editing of English language and style required

Author Response

Thank you for evaluating our manuscript. Thank you for your contribution and interest.

Changes were made based on your suggestions.

A native speaker has reviewed and corrected the text.

Some explanations were added to the methods and discussion sections.

Round 2

Reviewer 1 Report

Thank you for making the chnages suggested in my review. 

Author Response

Thank you for evaluating our manuscript and your contribution.

Mehdi S. Mirsaeidi M.D., Professor & Chief, Division of Pulmonary, Critical Care and Sleep Medicine, who is a native speaker, has reviewed and corrected the text.

Reviewer 2 Report

Need minor corrections

Author Response

Thank you for evaluating our manuscript and your contribution.

Mehdi S. Mirsaeidi M.D., Professor & Chief, Division of Pulmonary, Critical Care and Sleep Medicine, who is a native speaker, has reviewed and corrected the text.

Is the research design appropriate? Revised according to your recommendations.

Are the methods adequately described? Yes, they are described as you recommend.

Are the results clearly presented? Yes, they are explained as you recommend.

Are the conclusions supported by the results? Yes, they are revised according to your recommendations.